# Exploring the Distribution of Gardens in Suzhou City in the Qianlong Period through a Space Syntax Approach

Jiayan Yun , Wenbo Yu and Hao Wang *

College of Landscape Architecture, Nanjing Forestry University, 159 Longpan Rd., Nanjing 210037, China;
yunjiayan@snu.ac.kr or yun7@njfu.edu.cn (J.Y.); yuwenbo@njfu.edu.cn (W.Y.)
* Correspondence: wh9816@126.com; Tel.: +86-138-05161-9757

**Abstract:** This study explored the spatial distribution of Suzhou gardens in the Qianlong period (AD 1736–1796) through an innovative method combining spatial syntax and historical textual analysis. Through a spatial syntax approach, a street axis model analysis suggested that the greater the degree of integration and prosperity of a street, the denser the distribution of gardens surrounding it. A canal axis model analysis indicated that more gardens were built around canals that were less integrated and had less traffic. The accuracy and reliability of the axis model analysis results were validated using historical documents and images. The following was observed: (1) Gardens were densely distributed in the prosperous commercial areas at the northern and southern ends of Suzhou City. The more prosperous the location of a garden, the greater the influence on its popularity. (2) Gardens were concentrated in residential areas with booming businesses. (3) Large numbers of gardens were built along canals with poor traffic functions. Building gardens by diverting water into them reflected the complementary relationship between urban geographical characteristics and garden-building techniques. (4) Gardens promoted commercial development, and commerce drove the preservation and continuation of gardens. The findings revealed the complementary relationship between garden preservation and commercial development.

**Keywords:** Suzhou city; Qianlong period; classical gardens; garden distribution

## 1. Introduction

Since the late Ming Dynasty (AD 1368–1644), Suzhou City, the national cultural and trade centre at that time [1–4], had become a commercially prosperous and densely populated [5] city with high land prices [6–8]. By that time, garden culture had infiltrated all aspects of life [9–11]. Large areas of land in Suzhou City were covered by gardens [12]. There were more than 80 recorded gardens in existence in the Ming Dynasty, and more than 100 by the time of the Qing Dynasty (AD 1644–1912) [13], a majority of which were built during the reign of Emperor Qianlong [14]. The reign of Emperor Qianlong was another period of large-scale construction of Suzhou gardens after the late Ming Dynasty. The locations, scale, water sources and artistic merits of Suzhou gardens were deeply affected by urban development in the Qing Dynasty [15]. The intrinsic association between gardens and the city was an important part of urban landscape and life in Suzhou City [16].

From the Ming Dynasty to the Qing Dynasty, the spatial structure of Suzhou City also experienced some minor changes. There were about 500 streets in and around Suzhou City in the Ming Dynasty, whereas this number reached 612 during the reign of Emperor Qianlong. The mileage of navigable canals, however, had declined from 92 km in the late Ming Dynasty to 57 km in 1797 [17]. Did the change in urban spatial structure lead to the change in Suzhou garden distribution from the Ming Dynasty to the Qing Dynasty? As projects that took up large areas of land in Suzhou during the reign of Emperor Qianlong, what was the relationship between the spatial distribution characteristics of gardens and the larger urban spatial structure?

Over the past few years, this issue has been studied by several researchers. A study by Tiantian showed that the distribution of the gardens inside Suzhou City from the Tang Dynasty (AD 618–907) to the Qing Dynasty was subject to the influence of changes in the layout of political and commercial centres in Suzhou City. Tang and Song gardens are distributed primarily near the political centre, and Ming and Qing gardens are distributed mainly in commercial areas [18]. By comparing the Suzhou garden distribution maps of the Ming Dynasty and the Qing Dynasty, Jen-shu argued that the gardens were mostly located in the business districts inside the city in the Ming Dynasty. The change in commercial areas in the Qing Dynasty caused the difference in the spatial distribution of gardens between the two dynasties. For example, the Feng Gate area did not have many businesses in the Ming Dynasty, but it developed as an important area for business in the Qing Dynasty, becoming a concentrated garden area [19]. Jing found that new Suzhou gardens were concentrated at the edges of the northern and southern ends of the city in the Qing Dynasty, whereas the gardens in the town centre were mostly based on the old ones built in the former (Song and Ming) dynasties. She also argued that the change in Suzhou garden distribution in the Qing Dynasty was primarily the result of the change in urban functions. The political centre of Suzhou City shifted from the city centre area in the Ming Dynasty to the western part of the city in the Qing Dynasty, and the commercial centre area further expanded from the northwest part of Suzhou City in the Ming Dynasty to the northwest Changmen area in the Qing Dynasty, which promoted the gardens' expansion into that area [20]. Yong attributed the large-scale construction of Suzhou gardens during the Qing and Ming Dynasties to the rapid development of urban commerce [6].

These studies mostly compare the distribution of Suzhou gardens between several dynasties from a macroscopic perspective or attribute the major factor determining garden distribution to economic development. Few studies, however, have focused on the distribution of Suzhou gardens in only one historical period from a microscopic perspective. This study analysed the spatial distribution of Suzhou gardens during the reign of Emperor Qianlong using space syntax based on the streets, canals and garden plots shown in the *Map of Suzhou City*. First, we translated the *Map of Suzhou City* and plotted the map with scientific scales. We used space syntax to analyse the axis models of the streets and canals inside Suzhou City that bore important traffic functions, thereby identifying the overarching spatial structure of Suzhou City in that period. Next, we analysed the intrinsic connection of garden distribution to urban spatial structures based on the street and canal axis model calculation results in combination with the locations of Suzhou gardens. Lastly, we validated the connection between garden distribution and urban spatial structure using historical documents and images, based on which we summarized the distribution characteristics of Suzhou gardens.

Unlike the historical document- and image-analysis methods used in traditional historical research on cities and gardens, in this study, we integrated space syntax in urban history research to explore the distribution of Suzhou gardens. Through a research method that combines scientific spatial syntax methods with historical literature interpretation, we examined why Qing Dynasty gardens were distributed mainly in commercial districts. Previous studies have not thoroughly excavated the relationship between gardens and commercial development. A detailed analysis of the garden distribution characteristics of the Qianlong period will help establish an in-depth understanding of how Suzhou gardens evolved from the late Ming Dynasty to the Qing Dynasty and identify the geographic correlations of the changes in garden distribution during the transition from the Ming Dynasty to the Qing Dynasty. This research not only provides insights into the distribution of Suzhou gardens in the late Qing Dynasty but also reveals how Suzhou gardens were preserved throughout history.

## 2. Methods

### 2.1. Translation of the Map of Suzhou City

Drawn in 1745, the *Map of Suzhou City* is an ancient map of Suzhou City from the middle of the Qing Dynasty. It is the most complete map that has been preserved after the *Map of Pingjiang City* of the Song Dynasty (AD 960–1279) and the *Water Sources in Suzhou Prefecture* of the Ming Dynasty. The whole map is well scaled with accurate orientation and detailed labels. There are 612 streets and alleys, 128 temples, 28 ancestral halls and 25 government buildings, as well as more than 100 gardens and farmlands with text annotations in the map [21], which have provided a favourable condition for translating it into a practical historical map with scientific scales. As shown in Figure 1, gardens and farmlands were located at the southern and northern ends of Suzhou City; the industrial and commercial centre was situated in the northwest area near Chang Gate; the political centre in the southwest area was located near Xu Gate; the silk-weaving district was located in the northeast area near Lou Gate and the commercial district was located in the southeast area near Feng Gate [22].

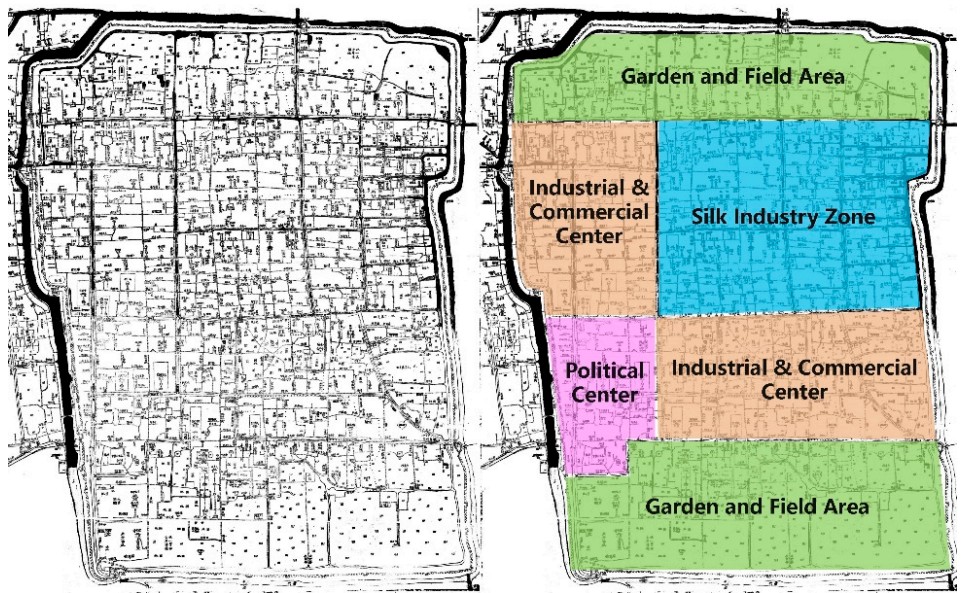

**Figure 1.** The *Map of Suzhou City* and its spatial distribution.

Yinong dedicated a significant portion of *The Chinese City in Space and Time* to demonstrate that even though ancient maps of Suzhou were not scientifically scaled or correctly plotted, the spatial topology of city walls and the water system they included were fairly accurate [23]. For this reason, it was still feasible to build the axis models using space syntax based on incorrectly scaled ancient maps [24]. Nonetheless, for our study, the *Map of Suzhou City* was translated with scientific scales to obtain more accurate research results.

We used the geographical information system (GIS) to retrieve the data of streets, canals and garden plots from the *Map of Suzhou City*. We then superimposed data on the current spatial layout of Suzhou City. Lastly, the historical information was positioned in the modern urban space of Suzhou City to complete the translation and plotting of the *Map of Suzhou City* (Figure 2) [25,26]. These operations were completed using ArcGIS10.2 software. As shown, the streets of Suzhou City during the reign of Emperor Qianlong were distributed with a checkerboard pattern and extended in all directions; the canals mostly ran along streets. Large garden plots and fields were situated near the city walls at the southern and northern ends of Suzhou City. Some smaller gardens and fields were scattered throughout the town centre.

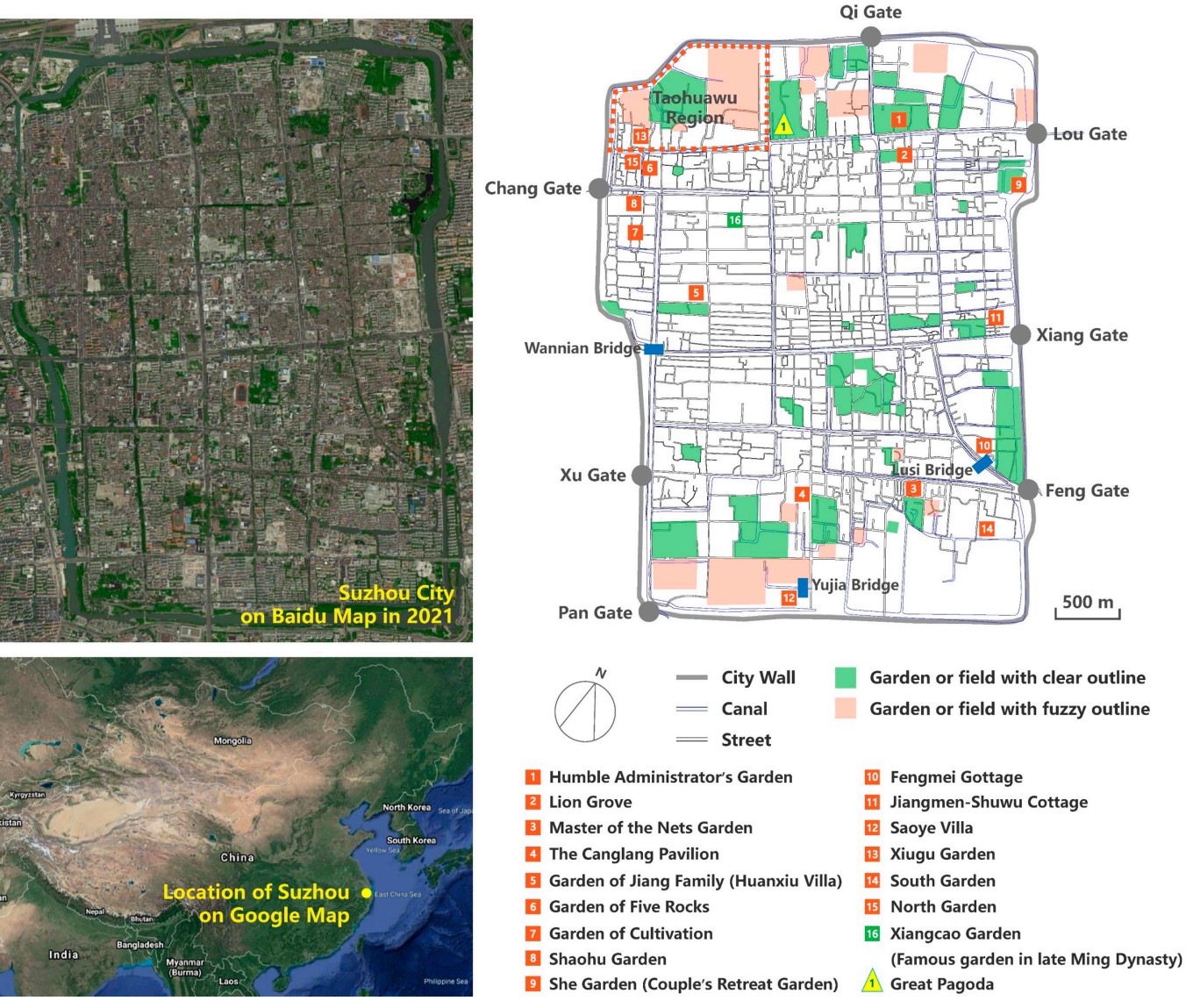

**Figure 2.** Translation of the *Map of Suzhou City* by scientific scales.

## 2.2. Axis Modelling and Analysis of Streets and Canals

Bill Hillier first proposed space syntax theory in 1974. He defined its core concept, "spatial configuration," in *Space Is the Machine* as follows: "if the spatial relationship between two spaces was defined as any form of connection between them, such as adjacency or interconnection, the initial relationship between them was subject to change when either of them was connected to a third space in any form, and there was configuration in it" [27,28]. Configuration means a set of relationships, any one of which depends on all other relationships related to it. As a spatial structure analysis method, space syntax provides a scientific and rational way to analyse urban space [29]. In studies that use space syntax theory, cities are simulated by generating the topological models of streets and amplifying various behavioural patterns [30]. Unlike the perceptual interpretation of space by traditional space analysis [31], the analysis of space syntax can identify the correlation between space form and space. As space form has feedback on and influences human activities, an inherent logical connection exists between form and function [32].

We manually plotted the axis map of streets and canals on the translated *Map of Suzhou City* with city walls as the boundary using the space syntax axis modelling method and AutoCAD. The axis map was then imported into DepthMap for calculation, which was followed by checking the Node Count values to validate the axis map and model accuracy.

After confirming the accuracy of axis plotting, we calculated the integration of street and canal axis models. The degree of integration refers to the degree of agglomeration or dispersion between a certain element and other elements in a spatial system. It measures the ability of space as a destination to attract arriving traffic and reflects the centrality of the space in the entire spatial system. The greater the number of intersections with other streets, the higher the degree of integration of the street, and the same is true for canals. The greater the number of intersections with other canals, the higher the degree of integration. The more integrated a space is, the easier it is for people to gather, which means that a more integrated space is more accessible [33]. Based on the principle that the more integrated an area is, the more accessible it is and the more cultural, economic and political functions it can provide [33], we analysed the spatial attributes of the areas where urban gardens were distributed to identify the spatial distribution of Suzhou gardens during the reign of Emperor Qianlong.

## 3. Results

### 3.1. Street Axis Model Analysis

The street axis model analysis (Figure 3) indicated that Xibei Street and Shiquan Street, located at the northern and southern ends of the city near the ancient city walls, had the highest integration degree in Suzhou City. That is, Xibei Street and Shiquan Street were the streets that had the greatest number of intersections with other streets throughout the entire Suzhou City. People could reach Xibei Street and Shiquan Street from multiple streets, which further demonstrated that these two roads had the greatest degree of accessibility for people and thus were the places where people gathered in the city. Xibei Street in the north extended eastward to Lou Gate. According to Cheng Zhanghua, a Suzhou local during the reign of Emperor Qianlong, "The eastern region of Suzhou City is Louguan [Lou Gate] where shops are densely scattered like stars and merchants from all over the country gather" [34]. This description not only shows that the Lou Gate region was a key commercial area in the city but also indirectly confirms the analysis result—that is, Xibei Street was a highly accessible street. Shiquan Street joined Feng Gate to the east. Gu Gongxie, a scholar from Suzhou during the reign of Emperor Qianlong, once wrote, "Pan Gate and Feng Gate used to be desolate areas. In the early period of Emperor Qianlong, people would not even bother to take a look at the luxurious houses on sale there. However, now they find that they cannot afford them at all" [35]. This quote describes the rapidly rising land prices and frequent commercial activities in the Feng Gate area at that time [19] and also confirms the reliability of the analysis result—that is, Shiquan Street had the highest degree of integration.

According to the analysis result, large areas of gardens were located along one or two blocks adjacent to the north side of Xibei Street, and a few small block-shaped gardens were located on the south side. By contrast, multiple block-like gardens were located along one or two blocks adjacent to the south side of Shiquan Street, and some smaller ones were located on the north side. The streets in Suzhou are distributed in a checkerboard pattern, and the distribution and area of each block are relatively average. The north–south length of each block is about 150 m. Gardens were the most densely distributed in the area around Xibei Street in Suzhou City during the reign of Emperor Qianlong; the garden distribution around Shiquan Street ranked second to that around Xibei Street. Thus, it was evident that the gardens were concentrated near highly accessible commercial areas in Suzhou City in that period, which was consistent with the conclusions that Jen-shu and Tiantian reached: Suzhou gardens were mostly located in economically prosperous areas during the Ming and Qing Dynasties [36]. Furthermore, our analysis also corrected Jing's opinion on the major reason why Suzhou gardens were distributed in the fringe areas at the northern and southern ends of the city during the reign of Emperor Qianlong. She argued that it was because that was where there was plenty of land [20], but both areas were the core commercial districts in the city that had shortages of land and neither were outlying areas.

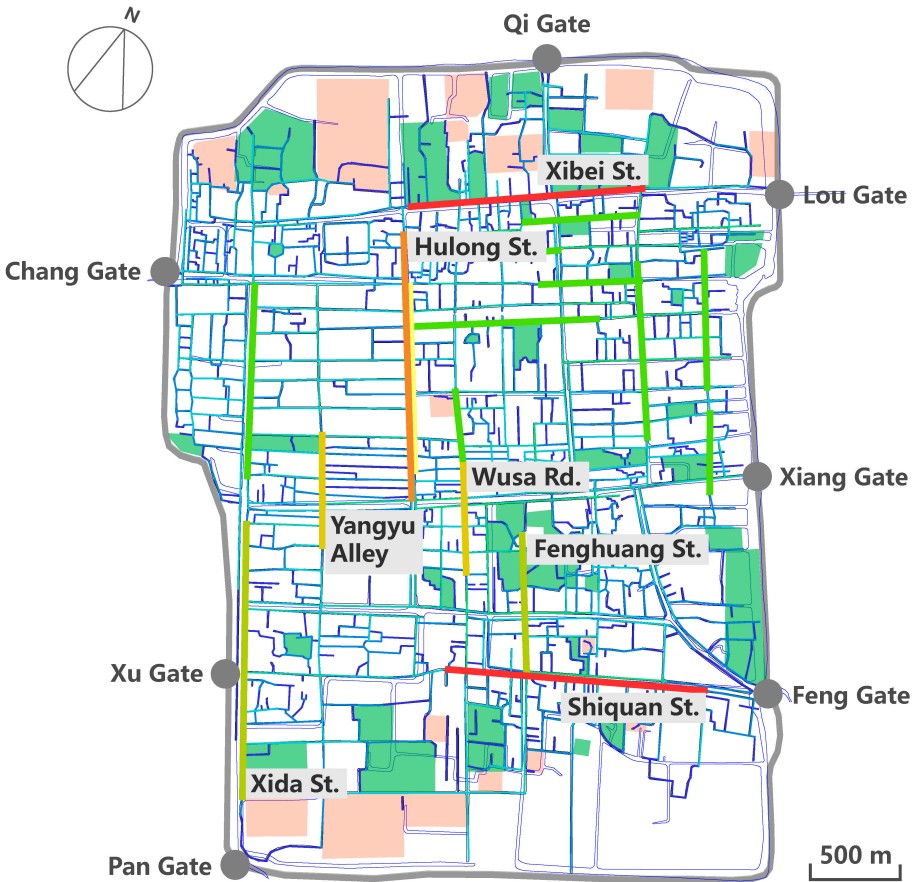

**Figure 3.** Street axis model analysis results. Streets marked in red are the most integrated streets, and those marked in dark blue are the least integrated streets. Streets with a higher degree of integration have more intersections with other streets, and streets with a lower degree of integration have fewer intersections with other streets.

After Xibei Street and Shiquan Street, Hulong Street had the third-highest degree of integration. Hulong Street connected Wu County, which was the western half of Suzhou City, and Changzhou County, which was the eastern half of the city. This street extended southward to the Confucius Temple and northward to the Great Pagoda. As a main road, it was the spine connecting a number of checkerboard-like blocks. However, few gardens were located along Hulong Street, suggesting that traffic arteries were seldom considered in garden site selection in Suzhou City.

Xida Street, Yangyu Alley, Wusa Road and Fenghuang Street were below Hulong Street in terms of integration, and a small number of gardens were located in one or two blocks around these streets. Xida Street stretched from Pan Gate in the south and ran through bustling marketplaces [37]. Large gardens were located near Pan Gate. In the Ming and Qing Dynasties, Yangyu Alley and Daoqian Street, where local government offices were located, were collectively known as the Daoyang district. Yangyu Alley, along which was located all kinds of shops and stores, used to be the residential area of government officials [37]. Gardens also could be seen along the alley. Wusa Road was adjacent to Fenghuang Street, which joined Shiquan Street to the south. Most of the buildings along both of these blocks were residences and some gardens also were scattered along them. Based on this analysis, we concluded that commercially prosperous residential areas were another type of area with densely distributed gardens in Suzhou City.

As shown in Figure 3, the streets marked in light green in the street axis model analysis were blocks with an average degree of integration. They were residential areas around which a small number of gardens were scattered. The rest of the streets (marked in light blue and dark blue) not only had a poor degree of integration but also had fewer gardens

distributed around them. For example, the streets along the area on the north side of Xibei Street and the south side of Shiquan Street that extended to the city wall had a very low degree of integration. Although large green areas surrounded them, the *Map of Suzhou City* has shown that these green areas were all farmland (Figure 4). Only one or two blocks near Xibei Street and Shiquan Street had gardens. Suzhou gardens were distributed mostly at the southern and northern ends of the city and were developed in commercial and mixed residential–commercial areas during the peak of the Qing Dynasty. Few gardens were located in residential areas with only residential functions. Thus, we concluded that the distribution of Suzhou gardens was tightly connected to the economic and trade districts in the city during the reign of Emperor Qianlong. Gardens were more densely located in areas with busier commercial activities.

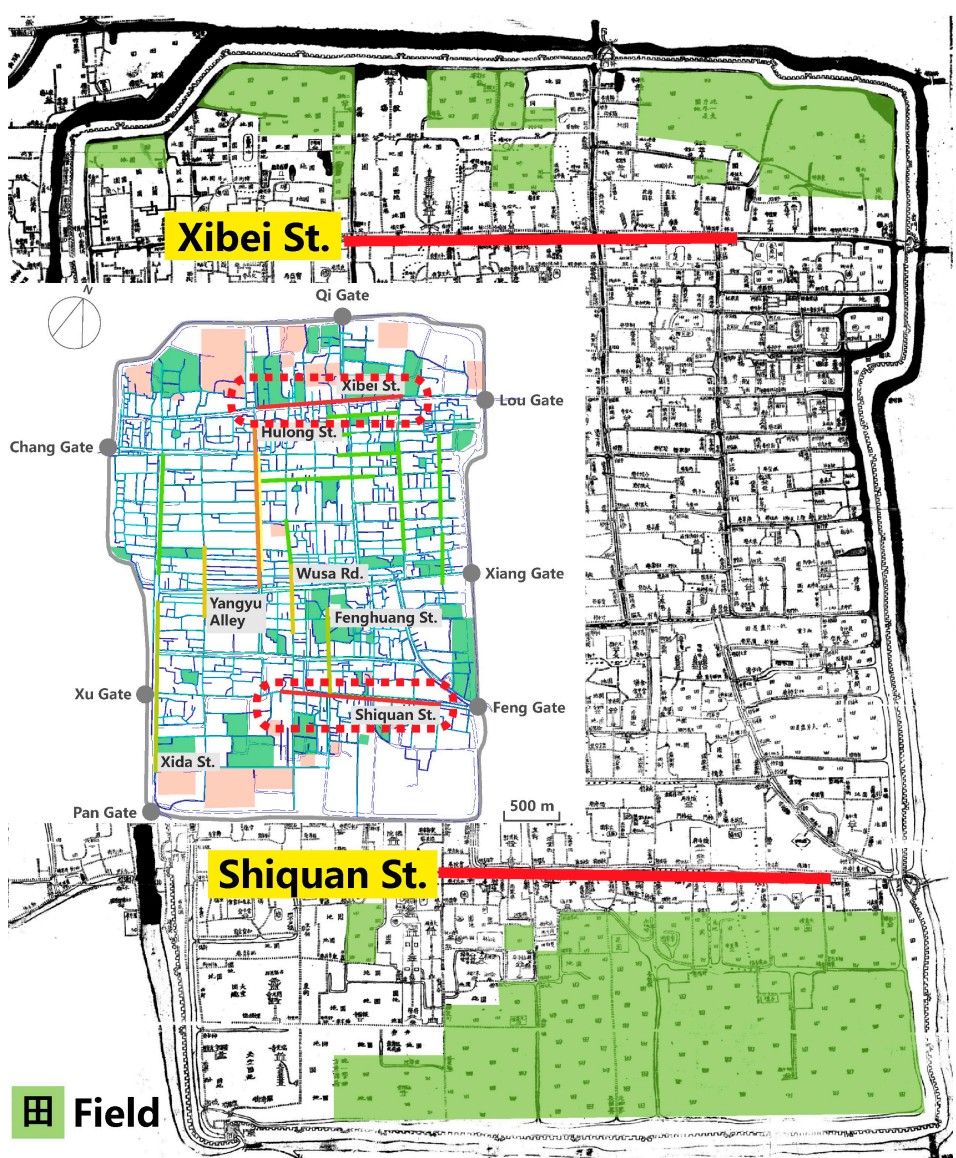

**Figure 4.** Fields on the north side of Xibei Street and the south side of Shiquan Street in the *Map of Suzhou City*.

## 3.2. Canal Axis Model Analysis

Canals were considered to be a major element of the urban landscape in Suzhou City [38]. Due to the existence of canals, streets were no longer solely responsible for transportation [39]. With the population boom in Suzhou City in the mid-Qing Dynasty [40–42], residences occupied a large amount of land, and many canals were either silted up or

polluted by domestic sewage [43]. Large numbers of canals were reclaimed as land in this context, which resulted in a sharp decrease in the number of canals [44], from a system of seven vertical canals and 14 horizontal canals in the late Ming Dynasty to one of six vertical canals and 10 horizontal canals [17]. In particular, the number of canals around Chang Gate dropped the most. In contrast, because the eastern city area near Lou Gate was the silk-weaving district, which required plenty of water [8,45], the number of canals there had been maintained and increased [17]. According to the comparison between canal distribution and garden distribution in the canal axis model analysis, several canals also were running through every garden either north–south or east–west, evidencing the close association between water sources and garden distribution.

The canal axis model analysis (Figure 5) indicated that the north–south canal marked in red, which flowed from Chang Gate to Pan Gate, showed the highest integration. This canal ran parallel to intermediately integrated Xida Street. There were a small number of garden plots distributed around it. The two canals marked in orange that ran north–south through the developed silk-weaving district near Lou Gate were the second most highly integrated canals, which shared similar garden distribution to the canal with the highest integration. The canals marked in yellow were intermediately integrated. They were located in the areas around Qi Gate, Lou Gate, Xiang Gate and the silk-weaving district. Gardens were more densely distributed along these canals than along those with higher integration. Canals marked in light green had average integration, but they were surrounded by more gardens. Despite their low integration, the canals marked in light blue and dark blue were surrounded by many small garden plots.

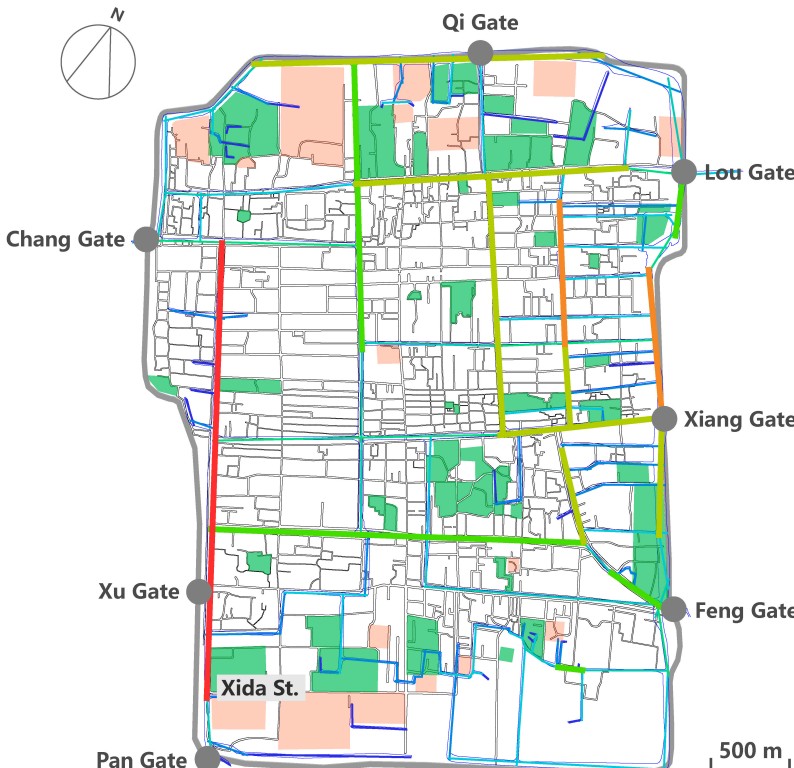

**Figure 5.** Canal axis model analysis results. Canals marked in red are the most integrated canals, and those marked in dark blue are the least integrated canals. Canals with a higher degree of integration have more intersections with other canals, and canals with a lower degree of integration have fewer intersections with other canals.

Based on this analysis, it can be inferred that fewer gardens surrounded the canals with a higher degree of integration and more gardens and denser gardens surrounded the canals with a lower degree of integration. That is, fewer gardens were built around

canals that had a greater number of crossings with other canals and higher accessibility. Canals played an important role in transportation in Suzhou City during the Ming and Qing Dynasties. There were stone carriers and cargo ships for logistic transportation, patrol ships used by the garrison and picket ships driven by farmers and sailors and private boats used by civilians as a means of transport [46]. Traffic was the most important function of urban canals. Fewer gardens could be seen around more heavily trafficked canals where there were more cargo ships, government ships and private ships, whereas gardens were denser around less-trafficked canals. This finding was consistent with the conclusion drawn by street axis model analysis—fewer gardens were distributed around Hulong Street, which featured stronger traffic functions. Although some green spaces were located at the southernmost end of the most integrated canal, according to the *Map of Suzhou City* (Figure 6), there were two temples on these green spaces. Another small piece of green space was located near the Wannian Bridge, but this is where the county government was located. There was also a small area of green space distributed around the canal with the degree of integration second only to the canal with the highest degree of integration. According to the *Map of Suzhou City*, however, this was a field (Figure 6). This also showed that the distribution of private gardens was indeed far away from the canals with a high degree of integration.

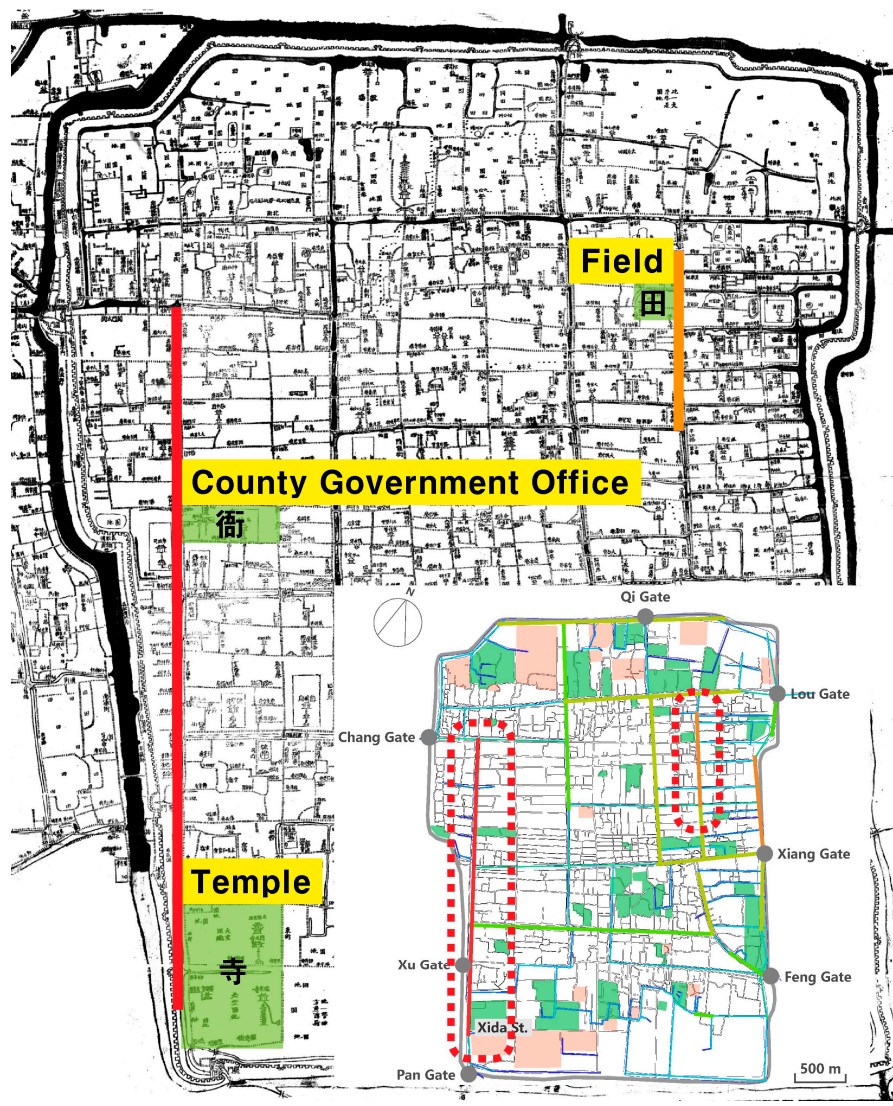

**Figure 6.** The distribution of green space around the canals with the highest and second-highest degrees of integration in the *Map of Suzhou City*.

## 4. Validation and Discussion

### 4.1. Gardens Were Densely Distributed in the Flourishing Commercial Areas at the Southern and Northern Ends of Suzhou City

The Humble Administrator's Garden was located along the north side of Xibei Street, which enjoyed the highest integration and accessibility in Suzhou City during the reign of Emperor Qianlong; the Lion Grove was situated along the south side. Located between Qi Gate and Lou Gate, the Humble Administrator's Garden was built by Wang Xianchen in the mid-Ming Dynasty, but it was possessed by the Xus in the late Ming Dynasty. During the early period of Emperor Qianlong, the western part of the garden belonged to a Mr. Ye, who served as a court historian, and it was renamed the "Shuyuan Garden"; the middle part of the garden was held by the Jiangs and was renamed the "Fuyuan Garden" [47]. Shen Deqian described the Fuyuan Garden as "a famous garden situated between Lou Gate and Qi Gate" [48], implying that it had gained a high profile. Equally famous was the adjoining Lion Grove, which was famed for its "springs and stones that were more spectacular than those in the whole prefecture" [48]. Thus, the Lion Grove was as popular as the Humble Administrator's Garden. The dense population and frequent commercial activities on Xibei Street further contributed to the popularity of both gardens. The finding that the most recognized gardens were situated by the streets with the highest accessibility confirmed the reliability of the street axis model analysis and the strong association between garden popularity and the spatial accessibility of the site where the famous gardens were located.

The Master of the Nets Garden and the Canglang Pavilion were located one or two blocks away to the south side of Shiquan Street, which was as accessible as Xibei Street. The Master of the Nets Garden, however, differed slightly from the Humble Administrator's Garden and the Lion Grove in location. According to *Notes of the Master of the Nets Garden*, "as a metropolis, the city proper of Suzhou has been filled with houses and crowded with people in nearly every part. However, in the southeast corner, there is a different place against the city wall beside a stream. It is covered by luxuriant vegetation, which makes it look like half-city, half-countryside" [49]. The Master of the Nets Garden was located "near the bustling marketplace, but people who visit here enjoy the tranquillity too much and forget to leave" [49]. Half of the Master of the Nets Garden neighboured the prosperous city and the other half abutted the countryside, which was consistent with the results of the street axis model analysis. Even though it was not rated as a "famous garden" or one "more spectacular than the whole prefecture", the Master of the Nets Garden was still an important garden in the period of Emperor Qianlong [50]. As it was situated only one block away from the highly accessible Shiquan Street, it was able to offer visitors a reclusive experience that contrasted with the bustling marketplace. As it was far away from the dense population, its popularity and influence were weakened.

The location of the Canglang Pavilion was different from that of the Humble Administrator's Garden, the Lion Grove or the Master of the Nets Garden. It was two blocks away from Shiquan Street and lied "across a block from the Master of the Nets Garden" [49]. Song Luo of the early Qing Dynasty once wrote, "the place near the Canglang Pavilion is untraversed. Wild streams are running freely across here. Giant stones are scattered everywhere. There are also many small hills among lush weeds" [51]. This area contained residences during the reign of Emperor Qianlong. By no means flourishing, at least it no longer looked deserted by that era (Figure 7). It was evident that although the Canglang Pavilion was near Shiquan Street, the two-block distance completely removed it from the dense population and boisterous streets. It still enjoyed a high profile because Emperor Qianlong visited it four times during his southern tour [52]. This further suggested that the influence of the most accessible commercial blocks in Suzhou City was confined to two blocks during the reign of Emperor Qianlong. Such spatial constraints could be broken by the visit of an influential figure, bringing popularity to the gardens that were not located in bustling commercial districts. In brief, the human factor played a role in enhancing the prominence of gardens that were confined by the urban spatial structure.

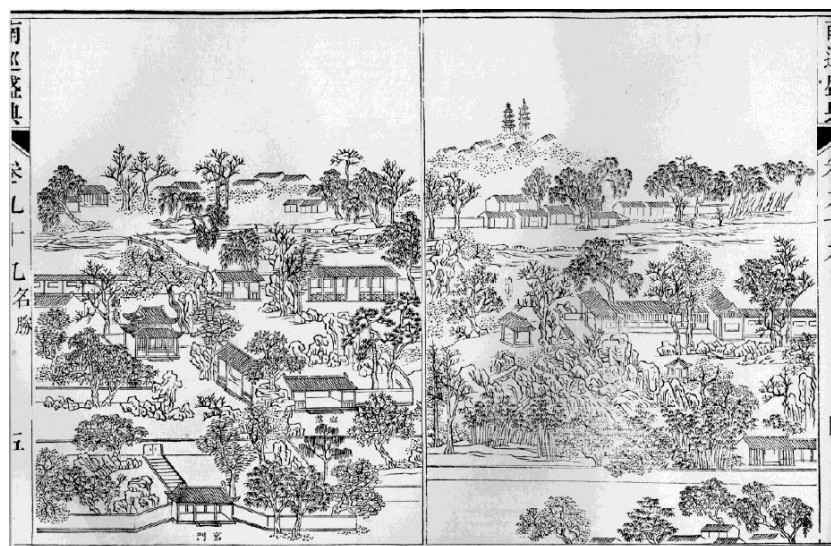

**Figure 7.** The Canglang Pavilion and its surroundings during the reign of Emperor Qianlong [53].

*4.2. Gardens Were Concentrated in the Commercially Prosperous Residential Areas inside Suzhou City*

With high integration, Yangyu Alley was a typical residential area for officials with a wide range of commercial functions in Suzhou City. The Garden of Jiang Family was built for Jiang Ji, who was an assistant minister of the Ministry of Penalty during the reign of Emperor Qianlong, at the northern end of Yangyu Alley, which extended in a north–south direction [54]. This garden was also known as the Huanxiu Villa of Secluded Beauty in the late Qing Dynasty. When building the garden, Jiang Ji invited Ge Yuliang from Changzhou to make rockeries. Jiang "built five houses, behind which small hills were piled using stones. He dug into the ground until clear spring water flowed down and the spring was named Flying Snow" [55]. Such splendid views led the garden to be rated as one of the "famous gardens in Suzhou City" until the end of the Qing Dynasty [56]. This description not only supports the popularity of gardens built around Yangyu Alley but also validates the results of the street axis model analysis. From the mid- to late Qing Dynasty, the spatial distribution of Suzhou City had not changed dramatically [57]. The Garden of Jiang Family, built in the mid-Qing Dynasty, remained a well-known garden. All of its views were preserved in the late Qing Dynasty, and it became part of the Huanxiu Villa, indicating that the preservation and continuation of a garden were closely associated with its location in the city.

Wenya Lane was connected to the intermediately integrated streets around Chang Gate, near which the Garden of Cultivation was located. Around the garden was "a commercial district that is boisterous all day long" [58]. Although it was a residential area for civilians rather than for officials, it was surrounded by frequent commercial activities (Figure 8). The Garden of Cultivation was said to be "lying among such bustling streets, known for the splendid views" [58]. Next to the Garden of Cultivation was the Garden of Five Rocks, which was located in Xihun Lane and connected to the intermediately integrated streets within the Chang Gate area. The garden was built in the late Ming Dynasty, but in the late Qing Dynasty, it was transformed into a private residence and held by the Shens [54]. Fortunately, the five rocks and the stone well had survived historical changes (Figure 9), evidencing that the garden had been properly preserved when it was a private residence. As famous gardens in the late Ming Dynasty and the early Qing Dynasty, both the Garden of Cultivation and the Garden of Five Rocks were well preserved, although they once served as private residences in the Qing Dynasty. The preservation reflected the enthusiasm for garden construction in this area. It also could be inferred that large numbers of gardens were located in the residential areas around the Chang Gate, and some of them were built by renovating or preserving old gardens of the former dynasty [14,19,54].

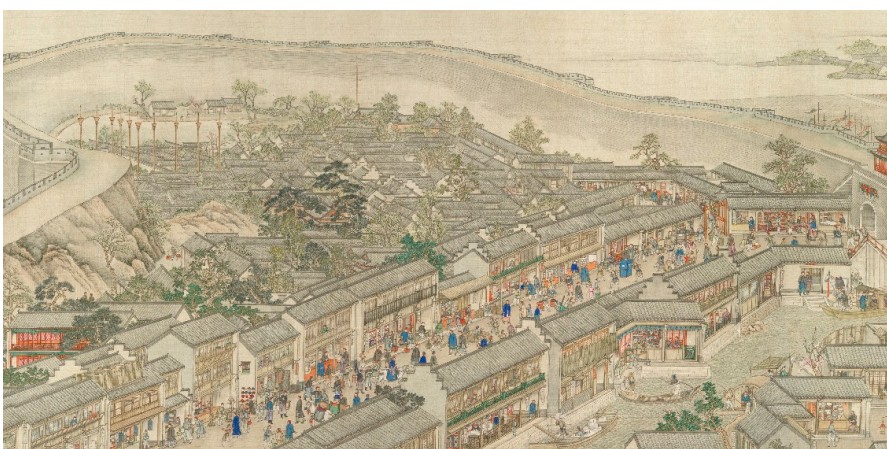

**Figure 8.** Residences and marketplaces within Chang Gate [59].

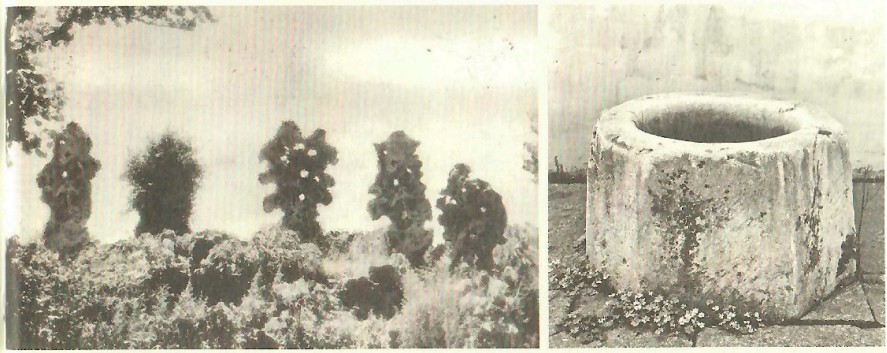

**Figure 9.** Photos of the five rocks and the ancient well in the Garden of Five Rocks [54].

In addition to the abovementioned two gardens, Shaohu Garden, a garden of the Fangs, was also located around the intermediately accessible streets within Chang Gate. In *Notes of Fangs Shaohu Garden*, Shen Deqian wrote: "Shaohu Garden is situated to the east of Pochu Gate. Due to the narrow space, the houses there are lined densely like fish scales to make the best of the land, which makes the place particularly clamorous. Amazingly, Shaohu Garden is a totally detached zone with elegance and tranquillity. People who pay their first visit do not even know there are bustling streets and crowded residential areas out of the garden. People who pass by have no idea of how fascinating the woods, stones, water, springs, birds and fishes are in the garden" [60]. Pochu Gate was another name for Chang Gate. Similar to the Garden of Cultivation and Garden of Five Rocks, Shaohu Garden was also located in a narrow downtown area. This description not only confirms the results of the canal axis model analysis but also demonstrates that new gardens were built in the residential areas with frequent commercial activities in Suzhou City during the reign of Emperor Qianlong.

The She Garden, also known as the Xiaoyulin Garden and renamed the Couple's Retreat Garden in the late Qing Dynasty, was located on Xiaoxinqiao Alley within the blocks with intermediate integration in the Lou Gate area. In *Notes of She Garden*, Cheng Zhanghua wrote: "Situated next to the eastern city god temple, the She Garden used to be the residence of Lu Jin who served as the chief of Baoning Prefecture. He normally lived in the old houses in complete seclusion and enjoyed burying himself in books" [34]. Cheng Zhanghua continued: "However, in the flowering season, he would open the gate and allow people to pay a visit" [34] to "remind them that there was a garden in such a bustling area" [34]. By doing so, the garden had become a semi-open tourist site for the public in Suzhou City. This opening implied that the area where the She Garden was located had a good flow of people, which was in line with the results of the street axis model analysis.

This also supported the phenomenon that private gardens were open to the public in the period of Emperor Qianlong.

*4.3. Gardens Were Scattered around the Canals with Low Integration*

Suzhou City took on a pattern of "streets and canals intertwining with each other" [61]. Most gardens were built by the canal. Fengmei Cottage, which was located by Lusi Bridge near the Feng Gate, once was the residence of Li Guo during the reign of Emperor Qianlong. Its gate faces the canal, with bridges over the river at both sides [62]. Situated within Xiang Gate, the Jiangmen-Shuwu Cottage was the residence of Zhang Dashou. As noted in *The Yuanhe County Annals of Qianlong Period*: "Lying by the canal, the garden is covered by thick ancient trees. Old branches spread in the sky, creating large areas of green shade and making the whole garden a tranquil place" [63]. The Saoye Villa was beside Yujia Bridge in the east of Pan Gate. In *Notes of Saoye Villa*, Shen Deqian noted: "It is built along a stream against the city, in which the trees were so luxuriant that the paths have been covered by fallen leaves. Therefore, people visiting the garden often get lost. Walking in it is like going into wild woods" [64]. All of these gardens were built by the canals with a poor degree of integration, which supports the results of the canal axis model analysis. In addition, the water nearby was diverted into the gardens by digging ditches, such as the case of the Master of the Nets Garden [61]. In *Notes of She Garden*, Cheng Zhanghua wrote: "Streams border on the three sides of the She Garden. Therefore, Mr. Lu drew water from them to the pools dug inside the garden" [34]. A mutually causal relationship existed between the checkerboard spatial pattern made up of intertwining streets and canals and the technique of diverting water into the garden. Such a relationship also reflected the influence of the geographic characteristics of urban space on garden-building techniques.

The canal with the highest integration in Suzhou City was the one that started from Chang Gate and ran to Pan Gate by way of Xu Gate. *Suzhou's Golden Age*, a realistic painting completed in the period of Emperor Qianlong, portrayed the scene around Wannian Bridge to the south of Xu Gate (Figure 10). In this painting, the varieties of and sheer number of ships on the canal, the frequent commercial activities and the large volume of people along it identify this canal as a major traffic artery. However, gardens were not found by the canal. As shown in Figure 11, many merchant ships, government ships, cargo ships and private ships could be seen sailing on the canal near Pan Gate; a large number of ships also were anchored off the bank. In particular, ships were densely packed at the junction between this canal and a smaller canal at the right side of the painting, suggesting that the canal must be a transportation hub in the city. Figure 11 also shows that gardens were not located along this canal section, but a few gardens were found around the noted smaller canal in the sections with less traffic. Once again, this finding supports the results of the canal axis model analysis and confirms the fact that gardens were built far away from major traffic arteries.

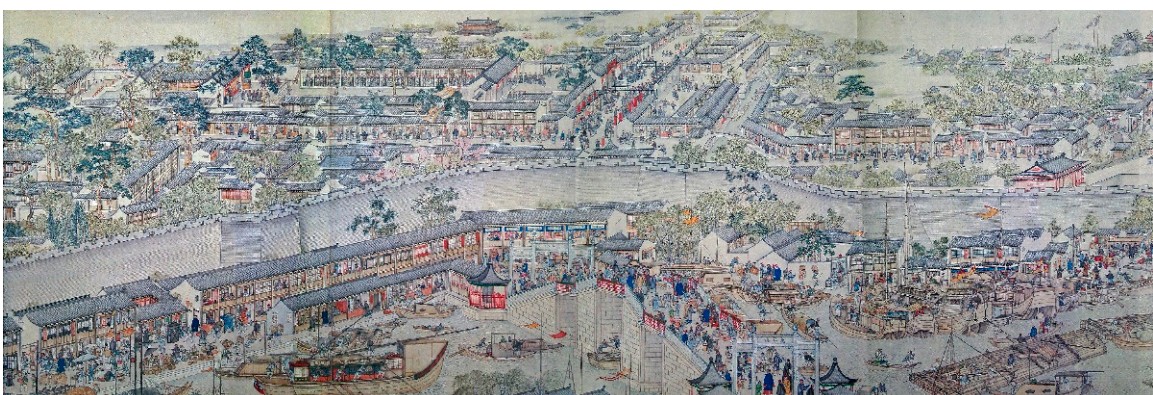

**Figure 10.** Scenes at Wangnian Bridge and the canals around it described in *Suzhou's Golden Age* [65].

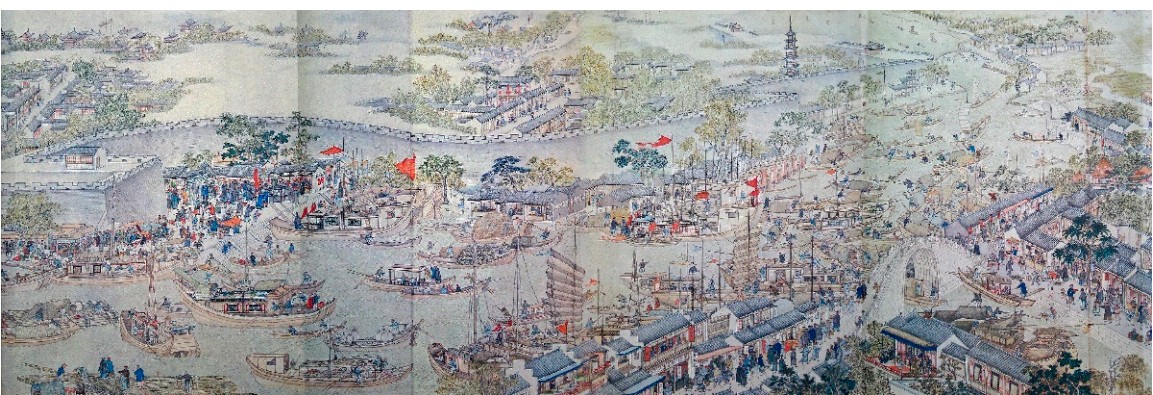

**Figure 11.** Scene of the canals around Pan Gate described in *Suzhou's Golden Age* [65].

*4.4. Gardens Promote Commercial Development; Commercial Activity Drives the Preservation and Continuity of Gardens*

In the late Ming Dynasty, the economically prosperous Peach Blossom Castle (Tao-huawu) area was once densely dotted with gardens [19], such as the Deserted Garden, the Lus' Citrus Grove, Hanya Village, the Small Peach Garden (Xiaotaoyuan), the Tangs' Garden (Tangjiayuan), the Sus' Garden (Sujiayuan), Mi'an Jiuzhu and the Wens' Hall of Magnolia [66]. During the Qing Dynasty, however, this area "was no longer prosperous and gradually became desolate. Looking from a distance, you will see hectares of fertile farmland lying there as flat as a palm" [67]. In other words, the Peach Blossom Castle area had been reduced to farmland during the reign of Emperor Qianlong [68]. A geographic analysis of the Peach Blossom Castle at the time showed that the streets in the area were poorly accessible. The number of streets was small, and most streets were broken roads that extended from the north to the south near the northern city wall. The number of intersections between these streets and other streets also was very small, and the accessibility was very poor. Apart from a highly integrated canal flowing in the east–west direction at the northern end, there were several canals with low integration in this area. The Xiugu Garden, which was built in the Qing Dynasty, also was located there. On the one hand, this fact validates the reliability of the street and canal axis model analysis; on the other hand, it highlights the important link between the decline of gardens in the Peach Blossom Castle area and the spatial change of Suzhou City during the transition from the Ming to the Qing Dynasties.

Another garden that was as famous as the Garden of Herbs (Yaopu Garden), the forerunner of the Garden of Cultivation, in the late Ming Dynasty was the Xiangcao Garden in Gaoshi Alley, which was rated as "a famous scenic spot located among bustling marketplaces" [69]. It was built by Wen Zhenheng, who was the author of *Superfluous Things*. Wen Zhenheng was the younger brother of Wen Zhenmeng, who was the owner of the Yaopu Garden [70]. Although this garden enjoyed popularity for a time in the late Ming Dynasty, it was ruined in the Qing Dynasty. An examination of the location of the Xiangcao Garden shows that Gaoshi Alley was in a poorly accessible area without any canals flowing through it during the reign of Emperor Qianlong. This location was in sharp contrast to the Garden of Cultivation, which was located in an area with both high accessibility and small canals. Once again, this finding demonstrates that the degree of development of the surroundings of a garden had an impact on its preservation.

Both the Humble Administrator's Garden and the Lion Grove, which were located in flourishing commercial areas and enjoyed a high profile during the reign of Emperor Qianlong, were built in the former dynasty, as were the Garden of Cultivation and the Garden of Five Rocks, which were situated in commercially prosperous residential areas. Instead of fading into history because of changing dynasties, they were preserved because of the developed business environment where they were located. This finding is at odds with the general perception that the more developed the commerce, the more buildings

and the less garden space. Therefore, Mote believes that the pressure of land use and the construction of gardens are intertwined to form the special spatial pattern of Suzhou City [71]. Mote has focused on the pure spiritual image expressed by Suzhou gardens as an ideal space, but in fact, these gardens also promote the prosperity and development of the surrounding economy.

Yuan Jinglan, a native of Suzhou in the Qing Dynasty, in his *Opening the Garden at Qingming Festival* records the Humble Administrator's Garden, Lion Forest and Canglang Pavilion as the leading gardens, and most of the gardens are open to visitors from the Qingming Festival to *lixia* (start of summer) [72]. Yuan noted: "In spring, the gardens open, and places for eating, lodging, and carriages are set up. Gardeners will ask visitors for money to enter the garden to enjoy the flowers. At that time, there will be a crowd of women, noisy carts and horses, and loud voices, making it too crowded to walk"; when the gardens opened, there were people selling candy, bait and small toys [72]. As Shen Fu described in *Six Chapters of a Floating Life*, written in 1793, he and his wife travelled to the South Garden near Canglang Pavilion and the North Garden in Chang Gate area when the rape blossomed [73]. Yuan Jinglan's *Appreciating Rapeseed Flowers in the North Garden and South Garden* noted that restaurants and tea houses were located everywhere to entertain tourists [74]. Qian Yong's *Lyuyuan Garden Collection* noted that in spring (February and March), when peach blossoms and rape blossoms bloomed, all the people in the city came out to visit the Lion Grove Garden, as seen in Zhang Zeduan's painting of *Along the River During the Qingming Festival* [68]. It also can be seen that during the Qianlong period, the gardens in the commercial area were not only separated from the commercial, public space as a private sector but also opened in the blooming season, making them a place for public leisure. Furthermore, they promoted commercial development and consumer culture around the garden. The entertainment functions and convenience facilities provided by the commercial premises also attracted a large number of visitors, which assisted in the preservation and continuation of the garden.

Craig Clunas believes that Suzhou gardens should be understood in the context of early modern China's commodity economy [10]. An analysis of the garden materials recorded in the book *Superfluous Things* in the late Ming Dynasty reveals that the Suzhou gardens are a reflection of the social economy and consumption patterns [2]. In the Qianlong period, the purpose of choosing exotic treasures to decorate the garden was no longer just to show the owner's elegant aesthetic taste and lofty social status as in the late Ming Dynasty [9], but rather to attract tourists into the garden. Yuan Jinglan's *Opening the Garden at Qingming Festival* wrote that to attract more visitors, "most gardens contain precious birds and exotic flowers. The buildings are hung with paintings and calligraphy works by famous literati, and decorated with tripods, pictures, books and antiques. Plant famous flowers in front of the steps inlaid with gold and jade. Set up colourful tents to shade the wind. Play music and sing traditional opera." [72]. Most of these garden decorations were sold in the commercial district of Suzhou, such as the Zhuanzhu Alley along the most integrated canal near the Chang Gate, where many shops sold antiques, rare books, ancient paintings and calligraphy, jade, gold, lacquer, carved wood and embroidery [2,75]. Such commercial blocks provide gardens with decorations that conform to the aesthetic trends of the times so that they can be renewed to keep up with the times. As a result, gardens became carriers that could promote the booming development of the commodity economy of Suzhou. This formed a model in which the preservation and continuation of the gardens and the prosperity and development of commerce complemented each other, and they prospered together.

## 5. Conclusions

In this study, we used space syntax to explore and analyse the spatial distribution of Suzhou gardens during the reign of Emperor Qianlong. Street axis model analysis showed that denser gardens were distributed around streets with a higher degree of integration and more frequent commercial activities, whereas fewer gardens were built

around streets with a lower degree of integration. The canal axis model analysis indicated that fewer gardens were distributed around canals with higher integration and stronger traffic functions, and more gardens were built around canals with lower integration and weaker traffic functions. We investigated historical documents and images to validate the accuracy and reliability of the results of the axis model analysis. Research findings suggest that the garden distribution in Suzhou City in the period of Emperor Qianlong showed the following characteristics: first, gardens were densely distributed in the commercially prosperous areas at the northern and southern ends of Suzhou City. The popularity of a garden was influenced by the prosperity of its locations. Second, gardens were concentrated in residential areas with booming businesses. Third, large numbers of gardens were built along canals with poor traffic functions. The technique of diverting water to build gardens reflected the complementary relationship between the urban geological characteristics and garden building techniques. Fourth, gardens promoted the development of Suzhou's commerce, and urban commerce also promoted the preservation and continuity of gardens. The study reveals that most of the gardens in the Qianlong Dynasty were transformed into semi-open public spaces, which attracted a large number of people and promoted the commercial development of the surrounding areas. The commercial districts also provided convenience and entertainment for tourists and thus attracted tourists to visit the gardens. Most of the objects sold in the commercial districts decorate the gardens, allowing the gardens to constantly adapt to the times and further promoting the preservation and continuity of the gardens. A complementary relationship existed between garden preservation and commercial development.

Today, some streets and canals in Suzhou City have changed to meet the needs of urban development. The old city, however, maintains its historical spatial pattern. As world cultural heritage sites, the Humble Administrator's Garden, Lion Grove, Master of the Nets Garden, Canglang Pavilion and Garden of Cultivation still stand in the city after years of change. Unlike the tradition that gardens were opened only in the blooming season in the Qianlong period, Suzhou gardens are now open to the public every day. Their long history and exquisite designs have attracted a large number of tourists. Due to these gardens, Suzhou City has become a notable tourist attraction, promoting the development of the urban economy and making gardens a symbol of the city's identity. Presently, the Humble Administrator's Garden and Lion Grove are still located on the prosperous commercial street, whereas the Master of the Nets Garden, Canglang Pavilion and Garden of Cultivation are quietly located among the dwellings, which is not much different from the Qianlong period. Although the position of Suzhou gardens in the urban space has not changed, because of historical transitions and urban development, their spatial attributes and social functions are constantly changing. In the future, as an important part of the contemporary Suzhou urban landscape, these gardens will continue to find new value as the urban landscape changes.

**Author Contributions:** Conceptualisation, J.Y.; methodology, J.Y.; software, J.Y. and W.Y.; validation, J.Y.; formal analysis, J.Y.; investigation, J.Y. and W.Y.; resources, J.Y.; data curation, J.Y.; writing—original draft preparation, J.Y.; writing—review and editing, J.Y.; visualization, J.Y.; supervision, H.W.; project administration, H.W.; funding acquisition, H.W. All authors have read and agreed to the published version of the manuscript.

**Funding:** This research was funded by the "National Key Research and Development Program of China" (grant number 2019YFD1100404) and "A Project Funded by the Priority Academic Program Development of Jiangsu Higher Education Institutions, PAPD."

**Institutional Review Board Statement:** Not applicable.

**Informed Consent Statement:** Not applicable.

**Data Availability Statement:** The data presented in this study are available on request from the author. The data are not publicly available due to [privacy]. Images employed for the study will be available online for readers.

**Conflicts of Interest:** The authors declare no conflict of interest.

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
