# Peer review of "Exploring the Distribution of Gardens in Suzhou City in the Qianlong Period through a Space Syntax Approach"

_land, doi:10.3390/land10060659_

Round 1

Reviewer 1 Report

The paper is very interesting and clearly written. The process is well described and appears well documented.
There are just a few suggestions.
The first is to clarify the concept of "accessibility" of roads. This is a focal element of the discussion and that should be made explicit: is accessibility in this paper a dimensional datum (width of the road) or does it identify some qualities of the roads themselves?
The second concerns the graphic quality of the maps, which deserve to be higher, both in terms of refinement and legibility (colors). An additional map would also be useful, placing the map of the historic center within modern and contemporary developments.
Finally, even if the approach of the paper is mainly historical, a treatment of this quality would also suggest some final consideration, on continuity or discontinuity in recent urban developments, on how such an important role of gardens in the formation of the historic city can represent a suggestion. for current developments.

Author Response

RESPONSES TO REVIEWER’S COMMENTS

Reviewer 1

Point 1: The first is to clarify the concept of "accessibility" of roads. This is a focal element of the discussion and that should be made explicit: is accessibility in this paper a dimensional datum (width of the road) or does it identify some qualities of the roads themselves?

Response 1: Accessibility refers to the degree of integration of streets, that is, the number of streets that are connected with a given street. The higher the degree of connection with other streets, the higher the degree of integration and the higher the accessibility. We included a detailed description of this in the last paragraph of Section 2.2, Axis Modelling and Analysis of Streets and Canals; please refer to lines 152-163. We also explained this use in the first paragraph of Section 3.1, Street Axis Model Analysis, and have made related modifications. Please refer to lines 166-172, 181-184, 201-202, 208-209. We also added related descriptions after Figure 3 and Figure 5; please refer to lines 221-224, 284-287. Corresponding changes also have been made in Section 3.2, Canal Axis Model Analysis; please refer to lines 271-273. We revised the title of Section 4.3, Gardens Were Scattered Around the Canals with Low Integration. For subtle changes, please see lines 413, 426, 466-468. We also adjusted the wording in 4.4. Gardens Promote Commercial Development; Commercial Activity Drives the Preservation and Continuity of Gardens and Section 5, Conclusions; please refer to lines 466-468, 545, 547.

Point 2: The second concerns the graphic quality of the maps, which deserve to be higher, both in terms of refinement and legibility (colors). An additional map would also be useful, placing the map of the historic center within modern and contemporary developments.

Response 2: We revised the illustrations in the paper, adjusted the colours and precision, and added a scale to the map, so that readers can more accurately understand the spatial scale of the city through the map. To ensure that readers can more accurately grasp the location of the city, we added a map of the geographic location of the historic location of the city of Suzhou in China as well as a map of the city of Suzhou today. Please refer to Figures 1, 2, 3, 4, and 5.

Point 3: Finally, even if the approach of the paper is mainly historical, a treatment of this quality would also suggest some final consideration, on continuity or discontinuity in recent urban developments, on how such an important role of gardens in the formation of the historic city can represent a suggestion for current developments.

Response 3: Thank you for the good advice. We have added this information to the last paragraph of the conclusion of the paper; please refer to lines 569-585.

Reviewer 2 Report

Strengths

  • Interesting but mostly because it shows the limits of the method

Weaknesses

  • Even if the paper approaches the spatial analysis of position of gardens it mostly stratifies/combines various descriptions that confirm each other, but it is difficult to understand how much of this knowledge could not be gained from a quick glance at the map.
  • The research lack clear definition of “close”, “distant”, “less”, “more” and so on making it difficult to understand whether the assumptions on “proximity” are actually true.

Generally:

  • The paper describes the analysis of garden distribution in Suzhou City based on the map from 1745 and additional historical documents.
  • The analysis is done using Space Synthax and historical text and graphic analysis.
  • The paper is interesting to read
  • The paper connects the level of integration of roads and channels with the distribution of the gardens. It concludes that most integrated roads had the gardens in proximity and that the most integrated channels had less gardens. It is also related to functions - commercial and residential of the roads.
  • The problem is that it doesn’t state anything new, nor does it help to understand why it is so. 
  • It finds the most integrated roads and channels as the most busy which was expected. And different types of activities that they are used for could directly explain the presence of gardens. The same goes for the level of building intensity…
  • Therefore the analysis is somehow circular - it uses the historic map, and the historic resources to describe the topics.
  • The papers is done relatively well, and is actually good example of the limits of the methods of spatial analysis.
  • The biggest problem is not clear definition of the what is considered as close and so on which makes it seems that those terms are used in a fluid way.
  • It is very obvious in analysis of the Xibei, Shiquan and Hulong streets that are more integrated than other roads, and Hulong st. crosses Xibei. Here it seems that for the distribution of green areas, some other factors were at play. Those are not taken into account or explained.  The less connected roads in eastern part of the city also have more green areas. Therefore it is difficult to explain this with axis analysis.
  • Also it is not clear what was the criteria for having “gardens near” something. At Xibei, the gardens touch Xibei street, at Shiquan not. Fenghuang st. passes through the garden and is less integrated. It is also not clear what having “more” or “fewer” green areas mean. It seems to be used too fluidly in the paper.
  • The similar comments stand for channel analysis as well. From the map it is not evident that there are more or less gardens at different points - for example at the channel near Xu Gate and at the channel between Xibei and Wusa road, but not only.
  • Historical documents describe the gardens and their use, but they do not confirm or explain the causal relation between the integration of roads or channels and gardens. 

In detail:

  • 85-86 - “space syntax has been introduced into urban history research for the first time in research on the distribution of Suzhou gardens” - it was actually first used in UK, London, by the authors of Space Synthax
  • 121-122 - “The GIS method was used to retrieve the data of streets, canals and garden plots from 121 the Map of Suzhou City” - it would be better to explain how, also what GIS software was used
  • 133-136 doesn’t have to be cited because it is easily available
  • 187-190 - it would be good to describe how “our analysis also corrected the opinion of Jing on the major reason why Suzhou gardens were distributed in the fringe areas at the north and south ends of the city during the reign of Emperor Qianlong.”

Author Response

RESPONSES TO REVIEWER’S COMMENTS

Reviewer 2

Point 1: The biggest problem is not clear definition of the what is considered as close and so on which makes it seems that those terms are used in a fluid way. It is very obvious in analysis of the Xibei, Shiquan and Hulong streets that are more integrated than other roads, and Hulong st. crosses Xibei. Here it seems that for the distribution of green areas, some other factors were at play. Those are not taken into account or explained. The less connected roads in eastern part of the city also have more green areas. Therefore it is difficult to explain this with axis analysis.

Response 1:

1. The degree of integration of streets is the number of streets that are connected with a given street: The higher the degree of connection with other streets, the higher the degree of integration and the higher the accessibility. We included a detailed description of this in the last paragraph of Section 2, Axis Modelling and Analysis of Streets and Canals; please refer to lines 152-163. We also explained this use in the first paragraph of Section 3.1, Street Axis Model Analysis, and have made related modifications. Please refer to lines 166-172, 181-184, 201-202, 208-209. We also added related descriptions after Figure 3 and Figure 5; please refer to lines 221-224, 284-287. Corresponding changes also have been made in Section 3.2, Canal Axis Model Analysis; please refer to lines 271-273. We revised the title of Section 4.3, Gardens Were Scattered Around the Canals with Low Integration. For subtle changes, please see lines 413, 426, 466-468. We also adjusted the wording in 4.4. Gardens Promote Commercial Development; Commercial Activity Drives the Preservation and Continuity of Gardens and Section 5, Conclusions; please refer to lines 466-468, 545, 547.

2. The streets on the north side of Xibei Street and the south side of Shiquan Street extended to the city wall gradually. The street integration is very low. Although there are large green areas around these streets, the Map of Suzhou City indicates that these green areas are all farmland (Figure 4). Only one or two blocks near Xibei Street and Shiquan Street actually have gardens. Please refer to Figure 4; lines 229-233.

Point 2: Also it is not clear what was the criteria for having “gardens near” something. At Xibei, the gardens touch Xibei street, at Shiquan not. Fenghuang st. passes through the garden and is less integrated. It is also not clear what having “more” or “fewer” green areas mean. It seems to be used too fluidly in the paper. The similar comments stand for channel analysis as well. From the map it is not evident that there are more or less gardens at different points - for example at the channel near Xu Gate and at the channel between Xibei and Wusa road, but not only.

Response 2: The standard for nearby gardens is that gardens are located within one or two blocks of highly integrated streets. We have given a more precise expression in the paper. Please refer to lines 185-189, 208-210, 232-233, 336. We also revised the annotations to the green areas marked in Figure 2, because some of these areas are gardens and some are fields. Please refer to Figure 2. For example, Xu Gate is surrounded by fields; please see the newly added Figure 4 and refer to Figure 4.

Point 3: Historical documents describe the gardens and their use, but they do not confirm or explain the causal relation between the integration of roads or channels and gardens. The paper connects the level of integration of roads and channels with the distribution of the gardens. It concludes that most integrated roads had the gardens in proximity and that the most integrated channels had less gardens. It is also related to functions - commercial and residential of the roads. The problem is that it doesn’t state anything new, nor does it help to understand why it is so.

Response 3: Indeed, the spatial syntax method has limitations. This article attempts to explain that the reason most gardens are located around river channels is for the use of water sources during gardening. See Section 4.3, Gardens Were Scattered Around the Canals with Low Integration, and the discussion in Section 4.4, Gardens Promote Commercial Development; Commercial Activity Drives the Preservation and Continuity of Gardens. We also explained why gardens are located primarily in commercial areas. Please refer to lines 413-453 and 454-541.

Point 4: 85-86 - “space syntax has been introduced into urban history research for the first time in research on the distribution of Suzhou gardens” - it was actually first used in UK, London, by the authors of Space Synthax.

Response 4: We revised these terms in the text; please refer to lines 84-86.

Point 5: 121-122 - “The GIS method was used to retrieve the data of streets, canals and garden plots from 121 the Map of Suzhou City” - it would be better to explain how, also what GIS software was used.

Response 5: We used ArcGIS 10.2 software. The specific operational steps have been indicated in the text. Please refer to lines 121-126.

Point 5: 133-136 doesn’t have to be cited because it is easily available.

Response 5: Thank you for your suggestion, but to maintain the scientific rigor, we opted to retain this citation..

Point 6: 187-190 - it would be good to describe how “our analysis also corrected the opinion of Jing on the major reason why Suzhou gardens were distributed in the fringe areas at the north and south ends of the city during the reign of Emperor Qianlong.”

Response 6: We provided a relevant description after this sentence; please refer to lines 182-184.

Reviewer 3 Report

Very nice paper.  This is a little bit out of my field but the writing and analysis is excellent.  I have two minor comments:

1) It would be useful somewhere to indicate the size of the area being studied, or even just adding a scale bar to one of the maps.  I didn't see where it is ever mentioned, so it's difficult for the reader to know how big of an area this is, which seems somewhat important.

2) Figures 5 and 6: can you indicate where these images came from?  Give a citation in the figure caption or otherwise let the reader know what the source was.  I didn't obviously see that.

thanks

Author Response

RESPONSES TO REVIEWER’S COMMENTS

Reviewer 3

Point 1: It would be useful somewhere to indicate the size of the area being studied, or even just adding a scale bar to one of the maps. I didn't see where it is ever mentioned, so it's difficult for the reader to know how big of an area this is, which seems somewhat important.

Response 1: Thank you for your suggestion. The scale has been marked in the figure. Please refer to Figure 2, Figure 3 and Figure 5.

Point 2: Figures 5 and 6: can you indicate where these images came from? Give a citation in the figure caption or otherwise let the reader know what the source was. I didn't obviously see.

Response 2: Thank you for your careful review. We added the source of the figures. Please refer to Figure 6, which is from the literature [53]; and Figure 7, which is from the literature [59].

Round 2

Reviewer 2 Report

The dimension of "close" is now expressed in blocks which is not precise information, please, define it in meters 

Author Response

Point 1: The dimension of "close" is now expressed in blocks which is not precise information, please, define it in meters

Response 1: The streets in Suzhou are distributed in a checkerboard pattern, and the distribution and area of each block are relatively average. The north-south length of each block is about 150 m. We have added a description in the text. Please refer to lines 189–192.
